# Elevated Bile Acid 3β,5α,6β-Trihydroxycholanoyl Glycine in a Subset of Adult Ataxias Including Niemann–Pick Type C

**DOI:** 10.3390/antiox13050561

**Published:** 2024-05-02

**Authors:** Nazgol Motamed-Gorji, Youssef Khalil, Cristina Gonzalez-Robles, Shamsher Khan, Philippa Mills, Hector Garcia-Moreno, Heather Ging, Ambreen Tariq, Peter T. Clayton, Paola Giunti

**Affiliations:** 1Ataxia Centre, Department of Clinical and Movement Neurosciences, UCL Queen Square Institute of Neurology, Queen Square, London WC1N 3BG, UKh.garcia-moreno@ucl.ac.uk (H.G.-M.);; 2Inborn Errors of Metabolism, Genetics and Genomic Medicine, UCL Great Ormond Street Institute of Child Health, 30 Guilford Street, London WC1N 1EH, UKp.mills@ucl.ac.uk (P.M.);

**Keywords:** biochemical biomarker, bile acid, Niemann–Pick type C, oxidative stress

## Abstract

Ataxia is a common neurological feature of Niemann–Pick disease type C (NPC). In this disease, unesterified cholesterol accumulates in lysosomes of the central nervous system and hepatic cells. Oxidation by reactive oxygen species produces oxysterols that can be metabolised to specific bile acids. These bile acids have been suggested as useful biomarkers to detect NPC. Concentrations of 3β,5α,6β-trihydroxycholanyl glycine (3β,5α,6β-triOH-Gly) and 3β,7β-dihydroxy-5-cholenyl glycine (3β,7β-diOH-Δ5-Gly) were measured in plasma of 184 adults with idiopathic ataxia. All patients were tested with whole genome sequencing containing hereditary ataxia panels, which include *NPC1* and *NPC2* mutations and other genetic causes of ataxia. Plasma 3β,5α,6β-triOH-Gly above normal (>90 nM) was found in 8 out of 184 patients. One patient was homozygous for the p.(Val1165Met) mutation in the *NPC1* gene. The remaining seven included one patient with Friedreich’s ataxia and three patients with autoimmune diseases. Oxidative stress is known to be increased in Friedreich’s ataxia and in autoimmune diseases. Therefore, this subset of patients possibly shares a common mechanism that determines the increase of this bile acid. In a large cohort of adults with ataxia, plasma 3β,5α,6β-triOH-Gly was able to detect the one patient in the cohort with NPC1 disease, but also detected oxidation of cholesterol by ROS in other disorders. Plasma 3β,7β-diOH-Δ5-Gly is not a potential biomarker for NPC1.

## 1. Introduction

Niemann–Pick disease type C (NPC) is an autosomal recessive lipid storage disorder. It is caused by mutations in the *NPC1* and *NPC2* genes, which lead to functional defects in proteins sharing the same names. NPC1 is a lysosomal transmembrane protein and NPC2 is an intralysosomal protein, and both play a crucial role in cholesterol efflux from the lysosomes [1,2,3]. In individuals with NPC, unesterified (i.e., the free, unmodified form) cholesterol accumulates in lysosomes, resulting in positive filipin staining in cells [4]. Subsequently, this unesterified cholesterol undergoes oxidation by reactive oxygen species (ROS), leading to the production and accumulation of oxysterols in lysosomes of the central nervous system cells and various other tissues throughout the body [5].

Cholesterol is an important constituent of cell membranes of eukaryotes. In plasma membranes, it accounts for 20–40 mol% of the total lipid and has a unique ability to increase lipid order while maintaining fluidity and diffusion rates [6]. It can protect other membrane lipids from attack by hydroxyl radicals without being degraded itself [7]. There are, however, circumstances in which it is attacked by reactive oxygen species itself. This has come to light, in particular, from studies of NPC disease.

In 2001, Alvelius et al. first identified unusual bile acids in the urine of an infant with NPC [8]. Subsequent papers have consistently confirmed that specific oxysterols—7-oxocholesterol and cholestane-3β,5α,6β-triol—are raised in the plasma of individuals with NPC [9,10], and it has become evident that bile acids produced through the metabolism of these oxysterols can be used as biomarkers for detecting individuals with NPC [11,12,13,14,15] (Figure 1).

There are two primary families of bile acids that can be employed as markers for NPC:(i)Those derived from 7-oxo cholesterol. These include 3β,7β-dihydroxy-5-cholenoic acid, which may be sulphated on the 3β hydroxyl group, conjugated with N-acetylglucosamine on the 7β hydroxyl group and conjugated with glycine (3β,7β-diOH-Δ5-Gly) or taurine at the C24 carboxyl group.(ii)Those derived from cholestane-3β,5α,6β-triol. These include 3β,5α,6β-trihydroxycholanoic acid, which may be conjugated with glycine (3β,5α,6β-triOH-Gly) and taurine [15].

The use of a bile acid conjugated with N-acetylglucosamine as a biomarker for NPC is limited by the fact that it will yield false-negative results in the 20% of individuals who carry an inactivating mutation in the gene encoding the GlcNAc transferase, *UGT3A1* [15]. An international panel convened to offer recommendations for the diagnosis of NPC pointed to 3β,5α,6β-triOH-Gly as one of the best biomarkers for detecting NPC [2]. This bile acid has been shown to be a promising biomarker for neonatal NPC screening [16].

Ninety-five percent of NPC patients have mutations in *NPC1*, with the remaining harbouring a mutation in *NPC2* [17]. NPC presents with a broad range of symptoms, ranging from a progressive fatal neonatal disorder to a milder form in adulthood that can evolve to chronic neurodegeneration. Progressive neurological defects are present in patients who exhibit the most common form of NPC, with a delay in motor development in early childhood, while in late childhood and in the juvenile form, symptoms manifest as abnormal gaits, ataxia, and cataplexy [2]. Ataxia is a common feature of NPC, but NPC is a rare cause of ataxia. In 2015, Synofzik et al. searched for mutations in *NPC1* and *NPC2* in 96 individuals with early-onset ataxia of unknown cause and detected four known *NPC1* mutations, three novel *NPC1* missense variants of uncertain significance (VUS), and one novel *NPC2* missense VUS. The total mutant allele frequency (8/192) was 4.17% [18].

Measurement of bile acids from both the 3β,5α,6β-triOH cholanoic acid family and the 3β,7β-dihydroxy-5-cholenoic acid family have proved useful in diagnosing NPC in asymptomatic neonates [12], in children aged 4 days to 19 years with predominantly neurological presentations [19], and in studies that included NPC patients over the whole age range from infancy to the 5th or 6th decade [13,14,15]. However, it was not clear whether 3β,5α,6β-triOH-Gly and 3β,7β-diOH-Δ5-Gly bile acids would yield accurate results in detecting NPC in adult individuals with ataxia. Therefore, in this study, we aimed to estimate the diagnostic accuracy of 3β,5α,6β-triOH-Gly and 3β,7β-diOH-Δ5-Gly as biomarkers of NPC in a cohort of adults with ataxia of unknown cause at the time of recruitment. Since the plasma concentration of any bile acid secreted by the liver can go up as a result of liver damage/cholestasis, we also measured the plasma concentrations of the normal bile acids, paying particular attention to cholyl glycine (glycocholate, 3α,7α,12α-trihydroxy-5β-cholanoyl-glycine), an isomer of 3β,5α,6β-triOH-Gly.

## 2. Methods

### 2.1. Study Population

All patients were recruited from the Ataxia Centre in the National Hospital for Neurology and Neurosurgery (NHNN) in London, UK. This centre is the main specialist referral centre for adult ataxia across the UK and other countries. The study was conducted from 2021 to 2023.

Patients referred to the Ataxia Centre outpatient clinic at NHNN were approached for consent to be included in the study if they had neurological features of ataxia and were aged >18 y. Informed written consent was obtained from all patients included in the study. The study was approved by the Bloomsbury Ethics Committee (Reference number: 95005) and was conducted in compliance with the Declaration of Helsinki.

In order to identify the sensitivity and specificity of the two bile acid biomarkers in a large cohort of adult ataxia patients with unknown cause, we used the multimodal approach, including the results of the analysis of bile acid biomarkers and the results of genetic tests in a large cohort of adult patients with unknown ataxia. These individuals were investigated using the standard assessment methods available in the ataxia clinic, including direct genetic testing, gene panel testing, performed at the Neurogenetic Laboratory at National Hospital for Neurology and Neurosurgery London UK, and the whole genome sequencing (WGS) as all the patients were recruited to participating in the 100,000 Genomes Project (https://www.genomicsengland.co.uk/initiatives/100000-genomes-project).

### 2.2. Bile Acid Analysis and Genetic Testing

Blood from the participants was taken into 12 mL EDTA tubes. Within 6 h, the plasma was separated by centrifugation at 3000 rpm for 5 min and divided into 1 mL aliquots, which were subsequently stored at −80 °C. The plasma samples were prepared for bile acids analysis using a previously published method [15]. Briefly, the plasma samples were defrosted and 10 µL were spotted onto neonatal screening cards (Whatman 903; Merck, Gillingham, UK) and left to dry overnight at room temperature. The whole spot was then punched out and placed in 300 µL of methanol containing 20 nM deuterated bile acids as internal standards. The samples were then sonicated for 15 min, eluting the bile acids from the spot. A calibration line was prepared for quantitation using the primary bile acids and their deuterated internal standards with a range of 0.1–250 nM. The samples were analysed by liquid chromatography–mass spectrometry using the same method previously reported [15]. Analysis was performed on a Waters ACQUITY UPLC coupled to a Xevo TQ-S triple quadrupole mass spectrometer with an electrospray ionization source in negative ion mode. A Waters ACQUITY UPLC™ BEH (Waters UK, Herts, UK) C18 column (1.7 μm, 2.1 × 50 mm) was used for chromatographic separation. Identification of 3β,5α,6β-triOH-Gly was based on retention time and the *m/z* 464 > 74 transition. These identifiers were confirmed by analysing the reference compound obtained from TLC Pharmaceutical Standards Ltd. (Newmarket, ON, Canada) (Appendix A). Only the glycine fragment ion of *m*/*z* 74 could be used for multiple reaction monitoring of *m*/*z* 464 (3β,5α,6β-triOH-Gly), as shown in the mass spectrum for *m*/*z* 464 fragment ions (Appendix A). Similarly, 3β,7β-diOH-Δ5-Gly was identified by its retention time and the *m/z* 446 > 74 transition. The previously determined cut-off for diagnosis of NPC from measurement of plasma 3β,5α,6β-triOH-Gly was 90 nM [15], and therefore, this level was considered as the cut-off point for detecting NPC in our study as well. The plasma level > 380 nM was determined as the cut-off point for 3β,7β-diOH-Δ5-Gly. The plasma bile acid assay was configured to also measure normal bile acids—cholic acid and its glycine and taurine conjugates and dihydroxycholanoates (unseparated chenodeoxycholic acid and deoxycholic acid) and their glycine and taurine conjugates. Cholyglycine has the same mass transition as 3β,5α,6β-triOH-Gly but a different retention time on the UPLC.

All patients underwent the whole genome sequencing (WGS) as part of the 100K Genomes Project (https://www.genomicsengland.co.uk/initiatives/100000-genomes-project), which contains a hereditary ataxia panel that includes the *NPC1* and *NPC2* genes and all the known genetic ataxias, including those due to repeat expansion [20].

### 2.3. Statistical Analysis

Statistical analysis was performed using STATA for Windows (version 14) and Microsoft Excel (2016). Analysis consisted of calculating sensitivity and specificity, along with creating a receiver operating characteristic (ROC) curve and area under the ROC (AUROC) curve.

## 3. Results

Overall, this study included 184 adult patients, initially presenting with unexplained ataxia. They ranged between the ages of 18 and 86 at the time of assessment. While all the patients underwent WGS testing, a final genetic diagnosis was achieved in 28 patients, one of them being diagnosed with NPC.

The median, range, and 96th centile values for plasma concentrations of 3β,5α,6β-triOH-Gly and 3β,7β-diOH-Δ5-Gly in the ataxia cohort are shown in Table 1. The cut-off for plasma 3β,5α,6β-triOH-Gly determined in our previous study (90 nM) was on the 96th centile for this cohort; the notes of the eight individuals with values above 90 nM were reviewed in detail. The 96th centile for plasma 3β,7β-diOH-Δ5-Gly in the ataxia cohort was 380 nM; the notes of the eight individuals with 3β,7β-diOH-Δ5-Gly > 380 nM were reviewed in detail. The plasma 3β,7β-diOH-Δ5-Gly and 3β,5α,6β-triOH-Gly concentrations are reported in Appendix A.

A scatter plot of bile acids 3β,5α,6β-triOH-Gly and 3β,7β-diOH-Δ5-Gly in our study can be found in the Appendix A. Overall, out of 184 patients, three had high 3β,7β-diOH-Δ5-Gly and 3β,5α,6β-triOH-Gly; five had high 3β,7β-diOH-Δ5-Gly only; and another five had high 3β,5α,6β-triOH-Gly only. The patient with NPC was among the latter group. Among these 13 patients, a genetic diagnosis was confirmed in four. Genetic diagnoses in the study cohort are reported in Table 2, and mutation details and bile acid levels are reported in Appendix A (found in Appendix A section).

The characteristics of individuals with high plasma 3β,5α,6β-triOH-Gly are presented in Table 3. All eight had normal plasma concentrations of cholylglycine and other normal conjugated bile acids and, thus, no evidence of liver disease/cholestasis. The patient with NPC was male and was 20 years old at the time of assessment, with an age of onset of 14. He had a plasma 3β,5α,6β-triOH-Gly level of 124 nM, and was homozygous for a known pathogenic mutation in *NPC1*-c.3493G>A; p.(Val1165Met). He was the only individual in the cohort with mutations in either *NPC1* or *NPC2*. The level of 3β,5α,6β-triOH-Gly was elevated in two other patients with genetic diagnoses—episodic ataxia type 2 (165.3 nM) and Friedreich’s ataxia (159.4 nM). However, there were four other individuals with episodic ataxia type 2 (*CACNA1* mutations) in the cohort and all of these individuals had normal plasma 3β,5α,6β-triOH-Gly (Appendix A). The plasma 3β,5α,6β-triOH-Gly was high in four individuals in whom a clear cause of ataxia was not determined. Three of these, numbers 51, 185, and 154 in Table 3, had an autoimmune disease. Patient No. 51 had diabetes, high anti-GAD antibodies, and a monoclonal gammopathy. Patients No. 185 and 154 had systemic lupus erythematosus (SLE) and idiopathic thrombocytopenia (ITP). Patients No. 51 and 185 also had elevated plasma 3β,7β-diOH-Δ5-Gly (450 nM and 964 nM, respectively). Patient No. 175 with no genetic diagnosis and no autoimmune features also had slightly elevated 3β,7β-diOH-Δ5-Gly (401 nM).

Of the five individuals with plasma 3β,7β-diOH-Δ5-Gly above 380 nM but normal (<90 nM) plasma 3β,5α,6β-triOH-Gly, four had no genetic diagnosis and no features that distinguished them from other members of the cohort without a genetic diagnosis (including no evidence of autoimmune disease). Only one individual had CANVAS syndrome with *RFC1* mutations. There was a second individual in the cohort with CANVAS syndrome with a normal plasma 3β,7β-diOH-Δ5-Gly (Appendix A). The median plasma concentration of 3β,7β-diOH-Δ5-Gly value for the cohort was 86 nM and the 95th centile was 334 nM. There were eight individuals with high plasma 3β,7β-diOH-Δ5-Gly. This included three individuals with very high levels (401, 450, and 964 nM) who were also in the eight who had elevated 3β,5α,6β-triOH-Gly—patients No. 51 (autoimmune features), No. 185 (SLE), and No. 175 (no additional features).

The contingency table for accuracy of 3β,5α,6β-triOH-Gly and 3β,7β-diOH-Δ5-Gly in diagnosis of NPC is demonstrated in Table 4. As it can be seen, the cut-off value of 90 nM for 3β,5α,6β-triOH-Gly had a sensitivity of 100% and specificity of 96.2% in detection of NPC, while the cut-off value of 380 nM for 3β,7β-diOH-Δ5-Gly had zero sensitivity.

Figure 2 also shows ROC curve analysis of accuracy of 3β,5α,6β-triOH-Gly in the diagnosis of NPC. This graph indicates that for 3β,5α,6β-triOH-Gly, a cut-off value of 122 nM yields a sensitivity of 100% and specificity of 96.2%. The area under the curve is 0.9836. As the 3β,7β-diOH-Δ5-Gly had zero sensitivity in the contingency table, a ROC curve was not applied for this conjugate.

## 4. Discussion

Historically, achieving a diagnosis of NPC has always been challenging. Histochemical examination of bone marrow for storage cells requires considerable expertise and the diagnosis could easily be missed. Filipin staining of cultured skin fibroblasts is labour-intensive and could not yield rapid results. Measurement of the oxysterols, cholestane-3β,5α,6β-triol, and 7-oxo-cholesterol in plasma have the disadvantage that ex vivo oxidation of cholesterol could lead to false-positive results. In 2017, an international committee of experts concluded that the most promising biomarkers for diagnosis of NPC were the bile acids, such as 3β,5α,6β-triOH-Gly and “lysosphingomyelin-509 (lyso-SM-509)”, which was subsequently identified as N-acylated and O-phosphocholine adducted serine [2,21,22].

Two studies suggested that plasma 3β,5α,6β-triOH-Gly was elevated in children with NPC [12,19]. Similar findings were yielded in a cohort of 73 individuals with genetically confirmed NPC1, ranging in age from 6 months to 54 years [15]. Therefore, this was the first of the two bile acids we chose to measure. In the family of bile acids derived from 7-oxo-cholesterol, many are sulphated and, so, rapidly excreted in the urine [14], some are conjugated with N-acetyl-glucosamine and, so, not produced by the 20% of individuals who lack an active GlcNAc transferase [15]; so, as a potential plasma marker in this family, we chose 3β,7β-diOH-Δ5-Gly. The purpose of this study was to determine the usefulness of these two bile acid biomarkers in detecting individuals with NPC in a very large adult cohort with unknown ataxia.

Evaluating plasma 3β,5α,6β-triOH-Gly as a biomarker for NPC in the ataxia cohort indicated that, using the previously determined cut-off of 90 nM, the one individual in the cohort with *NPC1* mutations was detected alongside seven other individuals (to be discussed below), so there was a sensitivity of 100% and specificity of 96.2%. Analysis of the ROC curve showed that a cut-off value of 122 nM produces a sensitivity of 100% and specificity of 98.4%, with an area under the curve of 0.984. We reviewed the ages of the NPC patients with plasma 3β,5α,6β-triOH-Gly < 122 nM in our previous study [15] and, although three were aged 3.2 y, 9.6 y, and 13.6 y (suggesting that, perhaps, levels may be lower in children), two NPC patients with plasma 3β,5α,6β-triOH-Gly < 122 nM were aged 26 y and 35 y. This suggests that using the cut-off of 122 nM might produce some false negative results; it is dangerous to set too much store by ROC curve analysis when there are few positive results, in this case only one. Indeed, the major limitation of this study is the fact that the cohort of 184 adults with idiopathic ataxia only contained one individual with NPC. In contrast, the study from Tubingen, Germany, identifying patients by sequencing *NPC1* and *NPC2* detected six cases in 204 individuals with early-onset ataxia [18]. It is possible that this difference is because our study was not limited to early-onset ataxia or because of a difference in the incidence of the disease between the UK and Germany. If we had detected five cases of NPC in our 184 adults with ataxia, the statistical analysis would have been more meaningful. However, we have been able to demonstrate that in adults with ataxia, measurement of plasma 3β,5α,6β-triOH-Gly can identify an individual with NPC with good sensitivity and acceptable specificity. Therefore, we have established that this bile acid is a good biomarker that predicts a positive genetic test in NPC1/NPC2.

In contrast, when results for plasma 3β,7β-diOH-Δ5-Gly were evaluated, it was clear that the individual with *NPC1* mutations had a value well within the normal range for the cohort (78 nM compared to the median of 85 nM) and this analyte could be of no value in screening for NPC. Previous studies have shown that increased amounts of sulphate conjugates of 3β,7β-dihydroxy-5-cholenoic acid are excreted in the urine of individuals with NPC [13]. This suggests that, while the urinary excretion of sulphate conjugates of 3β,7β-dihdroxy-5-cholenoic acid is determined mainly by the rate of oxidation of cholesterol to 7-oxocholesterol, the plasma concentration of 3β,7β-diOH-Δ5-Gly is affected more by other variables. Studies of inborn errors of bile acid synthesis (*CYP7A1* and *CYP7B1* mutations) have shown us that in most individuals, bile acids with a 3β-hydroxy-Δ5 structure are efficiently converted to sulphates and excreted in the urine, whereas in others, conversion to glycine and taurine conjugates predominate [23]. The addition of a glucosamine group to the 7β-hydroxy position of members of the 3β,7β-dihydroxy-5-cholenoic acid family of bile acids is completely blocked in the 20% of individuals that carry the inactivating mutation in the gene encoding the GlcNac transferase, *UGT3A1*. It seems likely that the major determinants of the plasma 3β,7β-diOH-Δ5-Gly are variants in sulfotransferase and GlcNac transferase enzyme activity that affect its further metabolism and not the rate of production of 7-oxo-cholesterol as a result of the reaction of ROS with cholesterol. Thus, it cannot be used as a biomarker for NPC. Interestingly two of the individuals with the highest plasma 3β,7β-diOH-Δ5-Gly also had elevation of normal bile acids, with no evidence of liver disease. This is of relevance because 3β,7α-dihydroxy-5-cholenoic acid inhibits the bile salt export pump [24], so 3β,7β-dihydroxy-5-cholenoic acid and its glycine conjugate may do the same.

Our study showed that some adults with ataxia who were shown to have no mutations in *NPC1* or *NPC2*, had high plasma concentrations of 3β,5α,6β-triOH-Gly. It behoves us to consider possible causes. Could there be a pathological process that increases the attack on cholesterol by ROS? Could there be reduced excretion of the bile acid(s) into bile? In the latter case, we might expect an increased plasma concentration of normal bile acids, including that of an isomer of 3β,5α,6β-triOH-Gly—3α,7α,12α-trihydroxycholanoyl glycine (glycocholate). All the individuals with elevated plasma 3β,5α,6β-triOH-Gly had a normal plasma glycocholate and normal values for all other conjugated bile acids, so cholestasis could be excluded as a cause of the raised plasma 3β,5α,6β-triOH-Gly.

For the individuals with raised plasma 3β,5α,6β-triOH-Gly and/or raised plasma 3β,7β-diOH-Δ5-Gly, having excluded cholestasis, we need to consider increased production of the abnormal bile acids as a result of increased attack of ROS on cholesterol.

Considering the individuals described in Table 3 (apart from Patient No. 20 with NPC): The first patient was patient No. 100, with a plasma 3β,5α,6β-triOH-Gly of 165 nM, who was heterozygous for a likely pathogenic mutation in *CACNA1A* [c.2636_2652dup p.(Ala885Thrfs*14)], indicating a diagnosis of episodic ataxia type 2 (EA2). We cannot really consider *CACNA1* mutations as a consistent cause of raised plasma 3β,5α,6β-triOH-Gly as there were four other individuals in the cohort with episodic ataxia who had normal plasma 3β,5α,6β-triOH-Gly and it is not clear exactly how a calcium channelopathy could lead to an increased attack on cholesterol by ROS.

Patients No. 185 and 154 with 3β,5α,6β-triOH-Gly of 100 and 219 nM, respectively, had strikingly similar additional features—a diagnosis of systemic lupus erythematosus (SLE) and thrombocytopenia—and an MRI scan showing progressive cerebellar atrophy and probable spinal cord atrophy. Cerebellar atrophy and spinal cord atrophy are both rare CNS manifestations of SLE [25,26]. The finding of increased bile acids derived from the action of ROS on cholesterol is compatible with the proposal that mitochondrial dysfunction resulting in oxidative stress is the cause of tissue and cell damage in T-cell-mediated autoimmune diseases, such as SLE. Evidence of peroxidation of other lipids is found in SLE [27].

Patient No. 51 with 3β,5α,6β-triOH-Gly of 109 nM also had evidence of autoimmune disease—Type 1 diabetes with anti-IA-2 and anti-GAD (glutamic acid decarboxylase) antibodies.

Patient No. 88 with 3β,5α,6β-triOH-Gly of 111 nM had additional features that were perhaps suggestive of a problem with elastase—emphysema and abdominal aortic aneurysm. Analysis of his 100K genome using Exomiser identified a complex variant in a splice region of the *CELA1* gene encoding an elastase involved in the pathogenesis of emphysema in α1-antitrypsin deficiency [28]. The same variant was present in patient No. 51. This might be suggesting that the *CELA1*-encoded elastase could act as a protease in antigen-presenting cells and when malfunctioning contributes to autoimmune disease. *CELA1* variants have been found in individuals with multiple autoimmune syndrome and Sjögren’s syndrome [29].

Patient No. 155 had a 3β,5α,6β-triOH-Gly of 159 nM. Analysis of the Frataxin gene (*FXN*) showed that the patient was compound heterozygous for two GAA expansions (length 75 and 101 bp), indicating a diagnosis of Friedreich’s ataxia (FRDA). This raises the question as to whether oxidation of cholesterol by ROS could play a role in the pathogenesis of FRDA, or at least, provide a marker of ROS damage. Studies in yeast have shown that when the frataxin homologue (Yfh1p) is deficient, iron builds up in the mitochondria and the yeast is susceptible to oxidative stress [30,31]. The combination of increased iron and acidity (as in the intermembrane space of the mitochondria) favours the production of ROS by Fenton reactions, but the cholesterol content of mitochondrial membranes is relatively low [32]. Pathological intramitochondrial iron deposition has also been described in conditional frataxin knockout mouse models and individuals with FRDA [33]. Transgenic FRDA mice-derived neurons show increased ROS production and lipid peroxidation [34]. This can be prevented by agents that prevent lipid peroxidation or activate antioxidant pathways within the mitochondria, providing good evidence for targeting these pathways as treatment for FRDA with a novel Nrf2 activator. This has been confirmed to be the only effective drug that has now been approved by the FDA for treatment of FRDA [35,36].

Patient No. 175 with 3β,5α,6β-triOH-Gly of 101 nM had no definitive genetic diagnosis, and unremarkable changes in cerebellum imaging.

Our results confirm previous reports that measurement of plasma 3β,5α,6β-triOH-Gly is a sensitive biomarker of NPC but measurement of plasma 3β,7β-diOH-Δ5-Gly is not, at least in the context of adult patients presenting with ataxia. However, our results also show clearly that increased plasma levels of bile acids that are produced by metabolism of oxysterols generated as a result of the action of ROS on cholesterol, are increased in other causes of ataxia, such as the disordered mitochondrial iron metabolism of FRDA. It is possible that the build-up of these bile acids is simply an indicator of increased ROS production, and it is the ROS that is causing the neuropathology; but it is also possible that intermediates on the pathway from 7-oxocholesterol and/or cholestane-3β,5α,6β-triol to the 24-carbon bile acids actually play a role in causing damage to cells of the CNS. The pathway from 7-oxocholesterol to 3β,7β-dihydroxy-5-cholenoic acid proceeds via the 27-carbon bile acid 3β,7β-dihydroxy-5-cholestenoic acid [15]. This bile acid is an agonist at LXRα and LXRβ receptors on neuronal cells, which, when activated, can promote the development of neurons and promote their survival [37]; however, it is less active than the cholestenoic acid produced by enzymatic oxidation of cholesterol—3β,7α-dihydroxy-5-cholestenoic acid—and, thus, may inhibit the action of this LXR activator.

## Figures and Tables

**Figure 1 antioxidants-13-00561-f001:**
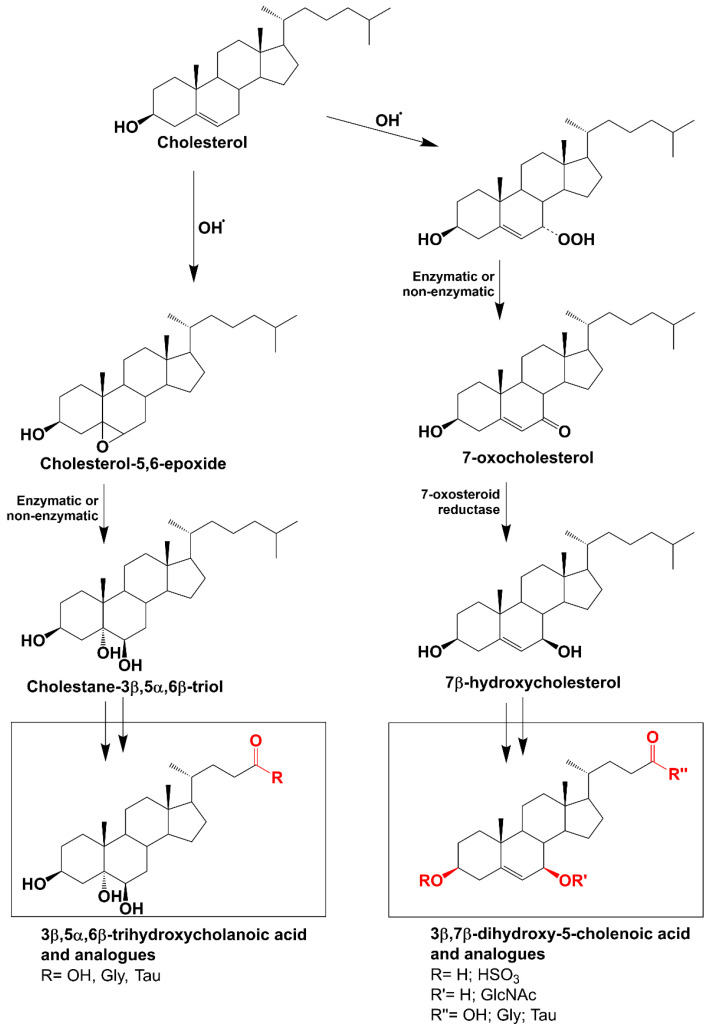
Oxidation of cholesterol by reactive oxygen species leads to the production of the oxysterols, 7-oxocholesterol and cholestane-3β,5α,6β-triol. Metabolism of these oxysterols leads to the production of two families of bile acids—3β,7β-dihydroxy-5-cholenoic acid and its conjugates and 3β,5α,6β-trihydroxycholanoic acid and its conjugates, respectively.

**Figure 2 antioxidants-13-00561-f002:**
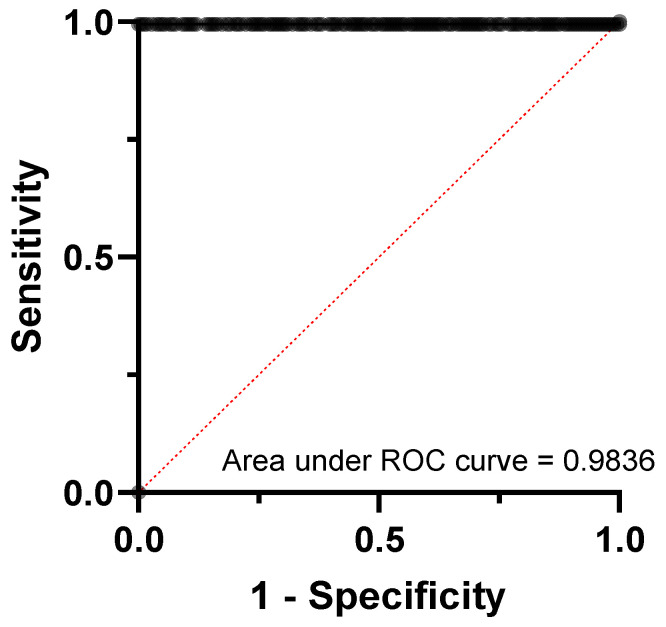
Receiver operating characteristic (ROC) curve depicting the accuracy of 3β,5α,6β-triOH-Gly in detecting NPC in adult patients with ataxia. Using a cut-off value of 122 nM has sensitivity of 100% and specificity of 96.2%.

**Table 1 antioxidants-13-00561-t001:** Plasma concentrations of 3β,5α,6β-triOH-Gly and 3β,7β-diOH-Δ5-Gly in the plasma of the 184 individuals in the cohort.

	Plasma Bile Acid Concentration nM (n = 184)	
	3β,5α,6β-triOH-Gly	3β,7β-diOH-Δ5-Gly
Median	22	85
Range	2–219	4–964
96th centile	90	380

**Table 2 antioxidants-13-00561-t002:** Characteristics of patients included in our study.

	Number	Age (Median; Q1, Q3)	Male/Female Number
**Total participants**	184	58 (43, 67)	93/91
**Patients with confirmed genetic diagnosis**	28	50 (35, 60)	16/12
*CACNA1* ^a^—related ataxia	5	40 (32, 56)	4/1
HSP ^b^	3	57 (52, 59)	3/0
CANVAS ^c^	2	71.5 (66, 77)	0/2
Leigh syndrome	2	61 (60, 62)	1/1
SCA ^d^-8	2	38 (36, 41)	2/0
*ANO10* ^e^—related ataxia	2	64 (57, 71)	0/2
**Niemann–Pick Disease Type C**	1	20	1/0
Friedreich’s ataxia	1	46	0/1
SCA-3	1	48	1/0
SCA-11	1	61	1/0
SCA-13	1	50	1/0
SCA-14	1	51	0/1
SCA-26	1	28	0/1
SCA-28	1	31	1/0
Charcot–Marie–Tooth Disease	1	27	1/0
AOA2 ^f^	1	34	0/1
Occult Macular Dystrophy	1	63	0/1
Mast syndrome	1	37	0/1
**Patients without genetic diagnosis**	156	59 (46, 68)	77/79

^a^ *CACNA1A*: calcium voltage-gated channel subunit alpha1 A gene. ^b^ HSP: hereditary spastic paraparesis. ^c^ CANVAS: cerebellar ataxia with neuropathy and vestibular areflexia syndrome. ^d^ SCA: spino-cerebellar ataxia. ^e^ *ANO10*: anoctamin 10 gene. ^f^ AOA2: ataxia with oculomotor apraxia type 2.

**Table 3 antioxidants-13-00561-t003:** Features of the patients with elevated plasma 3β,5α,6β-trihydroxycholanoyl glycine (3β,5α,6β-triOH-Gly).

	Pt n°20	Pt n°100	Pt n°51	Pt n°88	Pt n°185	Pt n°155	Pt n°175	Pt n°154
Sex	Male	Male	Female	Male	Female	Female	Female	Female
Age at sampling (yrs)	20	56	81	58	43	46	37	43
Bile acid 3β5α6β nM (0–90 nM)	124	165	109	111	100	159	101	219
Bile acid 3β7βΔ5 nM (0–380 nM)	78	83	450	59	964	105	401	127
Family History	-	+ (Autosomal dominant: sibling affected with same variant. Mother, sister, sister’s daughter, and another brother affected with similar condition, but not tested)	-	Unknown (adopted)	-	+ (Sibling also with biallelic GAA repeat expansions, 83 and 118)	-	-
Age of onset (yrs)	14	1 delayed milestones	10 epilepsy 64 nystagmus 68 ataxia	43 “shakiness” when bending down 56 ataxia	Delayed speech (14 months old: Speech and Language Therapy) 39 ataxia	33 ataxia, dysarthria	37 epilepsy, vertigo 40 ataxia, nystagmus	40 ataxia 46 dysarthria when tired 48 confusion, insomnia, depression
First sign	Ataxia	Ataxia	Gelastic epilepsy	“Shakiness” (tremor/imbalance)	Ataxia	Ataxia	-	Ataxia
Neurological Examination	Ataxia, dystonia, vertical gaze palsy	Ataxia, nystagmus, mild cognitive impairment	Ataxia, downbeat nystagmus	Functional signs (violent jerking movements of both legs)	Nystagmus, spasticity in all 4 limbs, ankle clonus, hyperreflexia, flexor plantars, ataxia? (antalgic gait, narrow-based)	Ataxia, spasticity, intention tremor, lower-limb hyperreflexia, bilateral upgoing plantars	Rotary nystagmus on left lateral gaze, downbeat nystagmus. Double vision on lateral gaze. Problems with speech, hearing and memory	Ataxia, dysarthria
MRI brain	Cerebellar atrophy	Cerebellar atrophy	Cerebellar atrophy	Cerebellar atrophy	Progressive (stable on last MRI) cerebellar and possible cord atrophy (none on last MRI), 2 brainstem telangiectasias	Unremarkable	-	Progressive cerebellar and possible cord atrophy
Additional conditions	Kartagener syndrome	Delayed milestones, learning disability, migraine	Epilepsy, diabetes, anti-GAD ^a^ but no insulin antibodies, monoclonal gammopathy	Functional neurological disorder, benign essential tremor, abdominal aortic aneurysm, adrenal adenoma, atrial fibrillation, emphysema, migraine	Spasticity, systemic lupus erythematous, depression, anxiety, lung cavity following severe pneumonia, thrombocytopenia treated with steroids at age 13, migraines, purple discolouration of her feet	No	-	Age 35 onset systemic lupus erythematosus resistant to immunosuppression. ANA and dsDNA positive ENA negative. Hypo complementaemia. Idiopathic thrombocytopenic purpura. Fibromyalgia
WGS ^b^	Homozygous for primary ciliary dyskinesia (*CCDC114*) & homozygous for p.(Val1165Met) mutation in *NPC1* gene	Heterozygous for c.2636_2652dup p.(Ala885Thrfs*14) likely pathogenic mutation in *CACNA1A* ^c^)	No primary finding (panels investigated: hereditary ataxia, brain channelopathy, familial hypercholesterolaemia, early-onset dyston ia)	No primary finding (panels investigated: inherited colorectal cancer (with or without polyposis), Parkinson’s disease and complex parkinsonism, hereditary ataxia, hereditary spastic paraplegia, thoracic aortic aneurysm or dissection). Also, DNA molecular analysis for cholestasis (24 genes): negative MLPA for whole exon deletions and duplications not performed	No primary finding (panels investigated: brain channelopathy, hereditary ataxia, mitochondrial disorders, hereditary spastic paraplegia, primary ovarian insufficiency, early-onset dystonia) familial hypercholesterolaemia, early-onset dystonia). Also, DNA molecular analysis for cholestasis (24 genes): negative MLPA for whole exon deletions and duplications not performed	Friedreich’s ataxia (compound heterozygous for 2 GAA repeat expansions: 75 and 101) (panels investigated: hereditary ataxia, brain channelopathy, hearing loss, hereditary spastic paraplegia) Previous to that, suspicion was *ANO10* ^d^ or *SPG7* ^e^ (Irish origin)	No primary finding (panels investigated: hereditary ataxia, epileptic encephalopathy, congenital disorders of glycosylation, undiagnosed metabolic disorders, mitochondrial disorders, early-onset dystonia, rare multisystem ciliopathy disorders, familial genetic generalised epilepsies) Also, whole mitochondrial genome sequencing: negative Also, next-generation sequencing (NGS) analysis of a panel of 11 genes involved in ion channel diseases of the brain and MLPA gene dosage analysis of 3 genes: negative	No primary finding (panels investigated: brain channelopathy, hereditary ataxia, mitochondrial disorders, hereditary spastic paraplegia, primary ovarian insufficiency, early-onset dystonia)
Final genetic diagnosis	Niemann–Pick disease type C	Episodic ataxia type 2	None	None	None	Friedreich’s ataxia	None	None

^a^ Anti-GAD antibodies: anti-glutamic acid decarboxylase antibodies. ^b^ WGS: whole genome sequencing. ^c^
*CACNA1A*: calcium voltage-gated channel subunit alpha1 A gene. ^d^
*ANO10*: anoctamin 10 gene. ^e^
*SPG7*: spastic paraplegia type 7 gene.

**Table 4 antioxidants-13-00561-t004:** Contingency table demonstrating the accuracy of raised 3β,5α,6β-triOH-Gly and 3β,7β-diOH-Δ5-Gly in the detection of Niemann–Pick disease type C (NPC).

	Genetic Diagnosis of NPC		
	Positive	Negative	Total
**Raised 3β,5α,6β-triOH-Gly (>90 nM)**			
Positive	1	7	8
Negative	0	176	176
Total	1	183	184
Sensitivity	100%		
Specificity	96.2%		
Positive predictive value (PPV)	12.5%		
Negative predictive value (NPV)	100%		
**Raised 3β,7β-diOH-Δ5-Gly (>380 nM)**			
Positive	0	8	8
Negative	1	175	176
Total	1	183	184
Sensitivity	0		
Specificity	95%		
Positive predictive value (PPV)	0		
Negative predictive value (NPV)	99.4%		

## Data Availability

Data are contained within the article.

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
