# Peer review of "Elevated Bile Acid 3β,5α,6β-Trihydroxycholanoyl Glycine in a Subset of Adult Ataxias Including Niemann–Pick Type C"

_antioxidants, 2024, doi:10.3390/antiox13050561_

Round 1
Reviewer 1 Report
Comments and Suggestions for Authors
The paper deals with an interesting validation study about a bile acid metabolite for the diagnosis of N-P C disease among adult patients affected by various types of ataxia. The validation is based on the analysis of a significant cohort (182 pts.), provided the disease at hand is a rare one.
Limitations are due to the fact that only a few patients, despite the good characterisation and the availability of a genetic diagnosis, have been used to validate the test.
Moreover, the metabolite is identified by just one mass-spec transition and a retention time. These are limited conditions but acceptable , in our view, provided that the authors already published another paper about the analytical validation of the test.
However, using at least two MS transitions could help improve specificity. Or at least, confirm analytical specificity by disclosing precedent validation of the single transition used.
Author Response
Dear Reviewer,
Many thanks for your valuable comments. I will address the points as below:
- Limitations are due to the fact that only a few patients, despite the good characterisation and the availability of a genetic diagnosis, have been used to validate the test.
We would argue that it is not possible to recruit very large numbers of individuals with ataxia of unknown cause over a 2-year period and a cohort of 184 was sufficient to contain one individual with NPC and 17 individuals with other genetic causes of ataxia.
- Moreover, the metabolite is identified by just one mass-spec transition and a retention time. These are limited conditions but acceptable, in our view, provided that the authors already published another paper about the analytical validation of the test. However, using at least two MS transitions could help improve specificity. Or at least, confirm analytical specificity by disclosing precedent validation of the single transition used.
We agree with the reviewer that it is best practice to have a qualifying transition in addition to the transition used for quantification in this type of analysis . However, the assay was optimised for maximum sensitivity for 3b,5a,6b-trihydroxy-cholanoyl glycine which is present at very low concentration in the plasma of controls. For these settings of cone voltage and collision energy, all glycine conjugated trihydroxycholanoic acids show only one fragmentation of appreciable size - that attributable to loss of the glycine with this moiety retaining the negative charge (464>74). Thus there is no qualifying transition available. When the study was carried out there was no commercially available reference compound but we now have the reference compound (and a deuterated version). We have used these to confirm that the retention time of the peak in plasma samples that we identified as 3b,5a,6b-trihydroxy-cholanoyl glycine has exactly the same retention time as the reference compound.
We have added this information to the manuscript, in materials and methods section line 133, highlighted in yellow:
"Identification of 3b,5a,6b-triOH-Gly was based on retention time and the 464>74 transition. These identifiers were confirmed by analysing the reference compound obtained from TLC Pharmaceutical Standards Ltd. (Ontario, Canada) (Figure S2). Only the glycine fragment ion of m/z 74 could be used for multiple reaction monitoring of m/z 464 (3b,5a,6b-triOH-Gly) as shown in the mass spectrum for m/z 464 fragment ions (Figure S3). "
Reviewer 2 Report
In this manuscript, the two bile acids were measured in 184 patients with ataxia to detect NPC. Compared to genetic diagnosis, the authors demonstrated that the plasma 3b, 5a, and 6b-triOH-Gly concentrations were good in diagnosing NPC disease. The authors discussed their results comprehensively. However, since all patients underwent whole genome sequencing, the NPC gene mutations can be identified without detecting plasma bile acids. In addition, the LC-MS method for bile acids detection was not easy.
There are two minor problems:
(1) On page 5, method 2.2 mentioned that "all patients underwent the whole genome sequencing." Why does the result say "28 received a final genetic diagnosis"?
(2) On page 14, patient No 6 had a 3b, 5a, and 6b-triOH-Gly of "109 nm", should be 159.
No
Author Response
Dear Reviewer,
Many thanks for your valuable comments. I will address the points as below:
- However, since all patients underwent whole genome sequencing, the NPC gene mutations can be identified without detecting plasma bile acids. In addition, the LC-MS method for bile acids detection was not easy.
Whole Genome Sequencing (WGS) is the most comprehensive form of genomic testing currently in clinical use. It enables a wide range of variant types in a large number of genes to be tested for simultaneously. Although it has many advantages, clinical interpretation of the large number of identified variants is a significant challenge, and many more variants of uncertain significance (VUS) are generated compared to more targeted testings. Furthermore, It is an expensive and time-consuming test. The bile acid 3b,5a,6b-triOH-Gly can be used as a functional biomarker for the identification of NPC condition as it can used as a relevant biomarker in determining the pathogenecity of a VUS.
That is why we are looking into techniques that can screen NPC much more quickly and with less cost.
- On page 5, method 2.2 mentioned that "all patients underwent the whole genome sequencing." Why does the result say "28 received a final genetic diagnosis"?
I apologize for not being clear. What we mean is that all 184 patients with idiopathic ataxia had the genetic tesing; however, only in 28 of them, a final genetic diagnosis was achieved. WGS was unable to find any primary genetic findings for the rest (156) of the patients.
I changed the result line 157 into this: While all the patients underwent WGS testing, a final genetic diagnosis was achieved in 28 patients, one of them being diagnosed with NPC.
- On page 14, patient No 6 had a 3b, 5a, and 6b-triOH-Gly of "109 nm", should be 159."
It was corrected. Thank you.
Reviewer 3 Report
The authors tested plasma levels of two oxidative byproducts of cholesterol (3β, 5α, 6β-triOH-Gly and 3β, 7β-diOH-Δ5-Gly) as potential biomarkers of Niemann-Pick disease type C (NPC). Overall, the paper provides acceptable background notions and scientific premise and context for the study.
However, I have a very big problem with this study: a population sample = 1 is not scientifically valid to draw any conclusion from the results shown here. Aside from the story statistics (ROC analysis) and reference values can tell, I am amazed by the fact that the authors state in their conclusions that 3β, 5α, 6β-triOH-Gly can discriminate between individuals with NPC and those without NPC. The authors have actually shown (Table 3) that at least 7 other individuals NOT AFFECTED by NPC have elevated levels of this bile acid, which alone proves that indeed the opposite of their conclusion is true. Actually, one could claim that elevated plasma levels of this bile acid may indicate disorders in which some kind of mitochondrial dysfunction is present (as also suggested by the row called WGS in Table 3, where "mitochondrial disorders" is highlighted in pretty much all the individuals), without any clear discrimination across disorders.
The text is overall pretty difficult to follow, especially since the individual IDs change from table to table. As an example, the same individual is identified as #20 in Table S1 and S2, and as patient n.1 in Table 3. In general, individuals should be indicated with the same ID throughout any given study.
Figure S1 legend does not provide enough details.
Author Response
Response to Reviewer 3
We note that Reviewer 3 has no criticisms of the scientific conduct of the research, only comments on presentation of results and particularly their interpretation and he/she questions whether the paper makes a relevant contribution.
With regard to the contribution, we would argue that there are two stages in evaluating a disease biomarker. In the first stage, results from a cohort of individuals affected by the disorder (in this case NPC) are compared to unaffected controls. If there is a good separation of affected individuals from controls, we can progress to the second stage which is to determine whether the test can be used in a particular clinical setting such as infants with liver disease or adults with ataxia. This paper is a report of a second stage biomarker evaluation. We believe it provides important information for the clinician seeing patients with undiagnosed ataxia; we have shown that, in this setting, measurement of plasma 3b,5a,6b-trihydroxy-cholanoyl glycine will detect a patient with NPC but there are other disorders that can also produce an elevated plasma level of this bile acid. The fact that there is evidence of cholesterol oxidation by ROS in these other disorders gives us important additional information about their pathogenesis.
Are the results presented clearly and in sufficient detail, are the conclusions supported by the results and are they put into context within the existing literature?
No
The three other reviewers answered “yes” to this question.
With an n = 1 it is an extremely long stretch to claim that "The plasma 3beta, 5 alpha, 6 beta-trihydroxycholanoyl-glycine demonstrated very good sensitivity and specificity in distinguishing between individuals with NPC1 and those without NPC1 disease".
We agree with the reviewer that we should be emphasising that there are disorders causing ataxia other than NPC that produce an elevated plasma 3b,5a,6b-trihydroxycholanoyl glycine.
We have changed the title to “Elevated plasma 3b,5a,6b-trihydroxycholanoyl glycine in Niemann Pick C disease but also in other disorders causing ataxia in adults”.
We agree with the reviewer that the “very good” the sentence he/she quotes overvalues the specificity of the test and the sentence has been removed. The previous sentence in the abstract gives values for the sensitivity and specificity that allow the reader to make their own evaluation. On Page 12 -13, in the two sentences starting with regardless of the exact cut-off…., we have changed “very good sensitivity and specificity” to “good sensitivity and acceptable specificity”. The section headed “Conclusions” also quotes very good sensitivity and specificity and this section has been removed.
Does this article provide a relevant contribution to the scientific discussion of this topic?
No
The other reviewers answered “yes” to this question.
At best, this study is just a confirmation (on an n = 1) of what has already been published.
As indicated above, this is a stage 2 evaluation of a potential biomarker. For a clinician seeing patients with undiagnosed ataxia, it is important to know what a positive result might mean; it could be NPC but there are other causes of ataxia that could give a positive result. This paper also shows for the first time that oxidation of cholesterol by reactive oxygen species is occurring in other disorders that cause ataxia. This may be very relevant to our understanding of the disease process.
Is it necessary to include study limitations in the discussion?
Yes
We have added this to the discussion after “It is dangerous to set too much store by ROC curve analysis when there a few positive results, in this case only one “:-
“Indeed, the major limitation of this study is the fact that the cohort of 184 adults with idiopathic ataxia only contained one individual with NPC. In contrast the study from Tubingen, Germany identifying patients by sequencing NPC1 and NPC2 detected 6 cases in 204 individuals with early onset ataxia [18]. It is possible that this difference is because our study was not limited to early onset ataxia or because of a difference in the incidence of the disease between the UK and Germany. If we had detected 5 cases of NPC in our 184 adults with ataxia, the statistical analysis would have been more meaningful.”
The authors tested plasma levels of two oxidative byproducts of cholesterol (3β, 5α, 6β-triOH-Gly and 3β, 7β-diOH-Δ5-Gly) as potential biomarkers of Niemann-Pick disease type C (NPC). Overall, the paper provides acceptable background notions and scientific premise and context for the study.
However, I have a very big problem with this study: a population sample = 1 is not scientifically valid to draw any conclusion from the results shown here. Aside from the story statistics (ROC analysis) and reference values can tell, I am amazed by the fact that the authors state in their conclusions that 3β, 5α, 6β-triOH-Gly can discriminate between individuals with NPC and those without NPC. The authors have actually shown (Table 3) that at least 7 other individuals NOT AFFECTED by NPC have elevated levels of this bile acid, which alone proves that indeed the opposite of their conclusion is true. Actually, one could claim that elevated plasma levels of this bile acid may indicate disorders in which some kind of mitochondrial dysfunction is present (as also suggested by the row called WGS in Table 3, where "mitochondrial disorders" is highlighted in pretty much all the individuals), without any clear discrimination across disorders.
In a cohort of 184 adults with ataxia of unknown cause at the time of referral, there was only one individual who was shown by the bile acid analysis and by sequencing of NPC1 to have NPC. As indicated above, from the German study we might have expected 5 and then the statistical analysis would have been more meaningful but we are reporting what we found. The cut-off for a normal plasma 3b,5a,6b-trihydroxycholanoyl glycine (94nM) was determined in our previous study using the same method of analysis so we could have included 75 individuals with proven NPC in our analysis which would have made the ROC analysis much more meaningful but the object of the study was to determine what happens when a measurement of plasma 3b,5a,6b-trihydroxycholanoyl glycine is made when a patient is referred with ataxia of unknown cause. We have reported the results accurately and, yes, there were 7 other individuals with elevated plasma concentrations of the bile acid who did not have NPC. This is what often happens with biomarker discovery; we though a-aminoadipic semialdehyde was a specific marker for pyridoxine dependent epilepsy – ALDH7A1 and then found it could also be positive in molybdenum cofactor deficiency and sulfite oxidase deficiency. We have changed the emphasis of the text to say that our results show that measurement of plasma 3b,5a,6b-trihydroxycholanoyl glycine can be used to detect individuals with NPC but that an elevated level is also seen in other causes of ataxia.
We agree that mitochondrial disorders can cause ataxia, hence they were included in the investigation of the patients including the tests indicated in the row headed WGS. However, these tests did not show any evidence of a primary mitochondrial disorder apart from the one case of Friedreich’s ataxia which can be regarded as a disorder of mitochondrial iron homeostasis. The discussion in relation to the 7 patients who had elevated plasma 3b,5a,6b-trihydroxycholanoyl glycine but did not have NPC considers the possibility of increased production of RPOS as a result of mitochondrial dysfunction but also considers other possibilities. We consider this is more balanced.
Detail comments
The text is overall pretty difficult to follow, especially since the individual IDs change from table to table. As an example, the same individual is identified as #20 in Table S1 and S2, and as patient n.1 in Table 3. In general, individuals should be indicated with the same ID throughout any given study.
Each participant in the study was allocated a number on recruitment. We have now changed the patient numbers in Table 3 to match the original participant number.
Figure S1 legend does not provide enough details.
We have added this legend: “Scatter plot illustrating the relationship between bile acids 3β,5α,6β-triOH-Gly and 3β,7β-diOH-Δ5-Gly, in our study. Each data point on the plot corresponds to a specific patient, with NPC patient being highlighted with an arrow. This patient had 3β,5α,6β-triOH-Gly of 123.9 nmol/L and 3β,7β-diOH-Δ5-Gly of 78.4 nmol/L.”
Reviewer 4 Report
In the article entitled « A bile acid produced through the action of reactive oxygen species on cholesterol detects Niemann-Pick C disease in adults with ataxia » (antioxydants_2844844), the authors' aim is to verify the relevance of a bile acid analysis for the diagnosis of NPC disease in a cohort of patients presenting with iodopathic ataxia of undetermined cause when recruited. Two bile acids were measured in blood samples, 3b,5a,6b-triOH-Gly and 3b,7b-diOH-D5-Gly. Such measurements of bile acids have previously proved useful in the diagnosis of NPC disease. Genomic analysis of a panel of genes involved in different pathologies characterised by ataxia complete the study.
Of the 184 patients studied, 8 showed an increase of the blood concentration of 3b,5a,6b-triOH-Gly (> 90nM), and genomic analysis revealed a mutation in the NPC1 gene in only one of them. The possible cause of the increased 3b,5a,6b-triOH-Gly concentration in the other 7 patients is discussed. The study shows also that the measurement of the plasma concentration of 3b,7b-diOH-D5-Gly could not serve as a biomarker for NPC disease associated with NPC1 mutation.
The scope of the study seems to be limited by the fact that there is only one patient with NPC in the cohort, but, of course, it is difficult to fault the authors for this ! Finding biomarkers is essential given the difficulty of establishing a rapid and reliable diagnosis, and this study undoubtedly provides new elements for the use of bile acid measurement in this respect.
The article is written in a clear manner and provides relevant information to the reader. Results are also clearly presented. I have no particular objection to the publication of this article in Antioxidants.
Minor point.
I'm wondering why Table S1 shows bile acid measured in 197 samples when the cohort includes 184 patients?
Author Response
Response to Reviewer No 4
We thank the reviewer for their positive comments. Our response to the negative comments are as follows:
Is it necessary to include study limitations in the discussion?
Yes
The scope of the study seems to be limited by the fact that there is only one patient with NPC in the cohort, but, of course, it is difficult to fault the authors for this !
We have added this to the discussion after “It is dangerous to set too much store by ROC curve analysis when there a few positive results, in this case only one “:-
“Indeed, the major limitation of this study is the fact that the cohort of 184 adults with idiopathic ataxia only contained one individual with NPC. In contrast the study from Tubingen, Germany identifying patients by sequencing NPC1 and NPC2 detected 6 cases in 204 indivduals with early onset ataxia [18]. It is possible that this difference is because our study was not limited to early onset ataxia or because of a difference in the incidence of the disease between the UK and Germany. If we had detected 5 cases of NPC in our 184 adults with ataxia, the statistical analysis would have been more meaningful.”
Minor point.
I'm wondering why Table S1 shows bile acid measured in 197 samples when the cohort includes 184 patients?
Potential participants were allocated a number. Thirteen either did not sign a consent form, or we did not obtain a suitable plasma sample, so these numbers are missing from Table S1: 45,59,84,101,102,109,128,133,147,157,180,195
Round 2
Reviewer 1 Report
The authors have convincingly addressed our previous criticisms in the updated form of the manuscript.
Although issues in mass spectrometry identification of proposed biomarkers were previously raised by us, the authors have provided all available methodological details adopted to increase confidence in its identification. These can be considered acceptable despite the fact that low resolution triple quadrupole mass spectrometry is not the MS best technique to be used to identify novel biomarkers.
In addition, despite the fact that we understand that the disease is a rare one, the number of positive cases examined is clearly very limited. This can be acceptable since the authors have clearly stated this obvious limitation in the text.
Author Response
Dear Reviewer 1, many thanks for your latest comment. We are very pleased you are totally satisfied with the recently amended version of the paper we have uploaded.
Reviewer 3 Report
I am really sorry, but the authors have not brought about convincing arguments that would make me change my mind about my previous opinions. The mere fact that the other 2 reviewers found the study acceptable and relevant, doesn't mean they're correct and this reviewer is not.
Once again, to draw scientifically sound conclusions about a given biomarker based on ONE subject with Niemann-Pick C disease is not valid or acceptable. Regrettably, unless the authors can confirm their findings on a bigger and more statistically sturdy subset my original position still stands.
I would like to clarify that my conclusions are not a reflection of the proficiency and expertise of the authors, who are obviously gifted and serious scientists. I just don't think this paper is convincing enough.
The authors have edited text and tables based on my criticisms. Nevertheless, in my humble opinion the main issue I brought up about this study still stands.
Author Response
We would like to thank the Reviewer 3 for his criticisms about the amended article we have submitted. We have taken them in to account and we have modified the title that is now: Elevated bile acid, 3β,5α,6β-trihydroxycholanoyl glycine in a subset of adult ataxias including Niemann Pick type C.
The abstract has been also now modified highlighting that the the elevation in bile acid is in a very low number of patients 8/184 that included patients with autoimmune diseases and Friedreich's ataxia and Nieman Pick type C cholesterole oxidation by ROS.
The conclusion has been modified mirroring the abstract conclusion.
I hope that all the amendments to the article are now satisfactory.